# First Report of Distinct *Bamboo mosaic virus* (BaMV) Isolates Infecting *Bambusa funghomii* in Vietnam and the Identification of a Highly Variable Region in the BaMV Genome

**DOI:** 10.3390/v14040698

**Published:** 2022-03-28

**Authors:** Ying-Wen Huang, Chin-Wei Lee, Na-Sheng Lin, Ha Viet Cuong, Chung-Chi Hu, Yau-Heiu Hsu

**Affiliations:** 1Graduate Institute of Biotechnology, National Chung Hsing University, 250 Kuo-Kuang Rd, Taichung 40227, Taiwan; ywhuang0101@gmail.com (Y.-W.H.); wei570211@yahoo.com.tw (C.-W.L.); 2Advanced Plant Biotechnology Center, National Chung Hsing University, Taichung 40227, Taiwan; 3Institute of Plant and Microbial Biology, Academia Sinica, Taipei 11529, Taiwan; nslin@gate.sinica.edu.tw; 4Department of Plant Pathology, Research Center for Tropic Plant Diseases, Vietnam National University of Agriculture, Hanoi 100915, Vietnam; hvcuongnh@vnua.edu.vn

**Keywords:** *Bamboo mosaic virus*, *Potexvirus*, satellite RNA, phylogenetic analysis, genetic marker, diversity, classification

## Abstract

New isolates of the *Bamboo mosaic virus* (BaMV) were identified in *Bambusa funghomii* bamboo in Vietnam. Sequence analyses revealed that the Vietnam isolates are distinct from all known BaMV strains, sharing the highest sequence identities (about 77%) with the Yoshi isolates reported in California, USA. Unique satellite RNAs were also found to be associated with the BaMV Vietnam isolates. A possible recombination event was detected in the genome of BaMV-VN2. A highly variable region was identified in the ORF1 gene, in between the methyl transferase domain and helicase domain. These results revealed the presence of unique BaMV isolates in an additional bamboo species in one more country, Vietnam, and provided evidence in support of the possible involvement of environmental or host factors in the diversification and evolution of BaMV.

## 1. Introduction

*Bamboo mosaic virus* (BaMV) is the main viral pathogen known to infect bamboos, posing serious threats to the bamboo industry. The presence of BaMV infections in different bamboo species in different plantations has been reported in many regions around the world, including Brazil [1], Taiwan [2], the United States [3,4], China [5,6,7,8], India [9,10], and Indonesia [11]. However, whether BaMV is also present in other major bamboo production areas or in additional bamboo species remained to be explored. The knowledge of the geographical or host distributions and the corresponding sequence variations may provide further insights into BaMV evolution and divergence.

BaMV is a member of the genus *Potexvirus*. The single-stranded, positive-sense RNA genome of BaMV contains a 5′-cap structure, a 3′-poly(A) tail, and five open reading frames (ORFs). ORF1 encodes the enzyme required for replication/transcription, consisting of three functional domains, namely the methyl-transferase, helicase, and RNA-dependent RNA polymerase [12]. ORFs 2–4 are organized into the “Triple-Gene-Block” (TGB) region [13], encoding three TGB proteins (designated TGBp1~3) involved in viral movement [14,15,16,17,18,19]. Specifically, TGBp1 may interact with viral RNA, replicase, and coat protein (CP) to form the viral replication complex (VRC), which is recruited by TGBp2- and TGBp3-containing vesicles to form the viral movement complex that moves through plasmodesmata (PD) into neighboring cells [16]. ORF5 is the coding region for CP responsible for virion formation, viral movement, replication, and symptom expression [19]. In addition, the 5′- and 3′-untranslated regions (5′-UTR and 3′-UTR, respectively) contain essential sequence or structural information for the replication/transcription of the viral genome [20,21]. Apart from BaMV genomic and subgenomic RNAs, satellite RNAs (satRNAs) have been shown to be associated with certain BaMV isolates identified in various regions of the world and co-evolve with the helper BaMV isolates [9,10,22]. The satRNAs associated with BaMV, designated satBaMV RNAs, are dependent on BaMV as the helper virus for replication and encapsidation and harbor an ORF for a P20 protein required for long-distance movement of satBaMV RNAs in host plants [23,24]. These ORFs and sequence motifs of BaMV and satBaMV RNA have been extensively characterized; however, whether these different regions are under similar selection pressure in their interactions with different host species in different geographical distributions remains to be investigated.

Vietnam is among the top 10 bamboo producers in the world, and the bamboo industry in Vietnam ranks fourth in the market as estimated in 2020 (https://www.inbar.int/international-bamboo-and-rattan-trade-key-takeaways; accessed on 15 June 2020). Previous phylo-geography studies have identified at least three main phylogenetic clusters of BaMV and satBaMV RNAs that reflected the geographical origins of the BaMV isolates [6,7,10]. However, the presence of BaMV or satBaMV RNA in bamboo plantations in Vietnam has not been officially documented previously. If BaMV is present in Vietnam, the analysis of phylogenetic relationships between Vietnam isolates and those from other geographical distributions in the world may provide further insight into the genetic variabilities and evolution of BaMV.

In this study, distinct BaMV isolates were identified from two bamboo leaf samples collected from an additional bamboo species in Vietnam, designated BaMV-VN1 and BaMV-VN2, for the first time to our knowledge. Unique satBaMV RNAs, namely satBaMV-VN1 and satBaMV-VN2, were also identified to be associated with BaMV-VN1 and BaMV-VN2, respectively. Sequence analyses were performed to examine the relationship between the Vietnam isolates and those reported in the other regions of the world. A highly variable region located in between the methyl-transferase and helicase motif of the ORF1 coding sequence of the BaMV genome was identified, which possibly reflects the interactions between BaMV and host defense systems. These results provide further hints on the effects of environment and/or host factors on the divergence of BaMV genomes.

## 2. Materials and Methods

### 2.1. Sample Collection

Two samples of leaves of bamboo (*Bambusa funghomii*), designated VN1 and VN2, that showed mosaic symptoms were collected from two different bamboo bushes in a bamboo plantation in Vietnam (Xuan Cam Commune, Hiep Hoa District; GPS coordinates:21.333000, 105.967000). The leaf samples were maintained as dried tissues in the presence of calcium chloride (CaCl_2_) powders and kept in a refrigerator at 4 °C.

### 2.2. Cloning, Sequencing, and Inoculation Assays

Total RNAs were extracted from bamboo leaf samples by using TriPure isolation reagent (Roche Life Science, Germany) following the instruction of the manufacturer. The extracted RNAs were used as the templates for 5′ rapid amplification of cDNA ends (5′-RACE) using primers derived from consensus sequences of BaMV (Ba652R: 5′-GTGTAGATTTCTGGGTATAG-3′), following the protocol of SMARTer^TM^. RACE cDNA Amplification Kit (Takara Bio, San Jose, CA, USA) to obtain the 5′-terminal sequences of the BaMV Vietnam isolates. Then, the 5′ primer (5′-GCGATATCGGAAAAGCAATCCAAACAAAC-3′) was synthesized based on the above information and used to amplify the full-length genome of the BaMV Vietnam isolates together with the Oligo-d(T) primer (5′-GCGATATCTCTAGATTTTTTTTTTTTTTTTTTTTT-3′) by reverse transcription-polymerase chain reaction as described previously [25]. The amplified full-length genomic cDNAs were then inserted into the *Stu*I site of the pCass vector [26] to generate infectious clones of the BaMV Vietnam isolates since the purified DNAs of pCass-based constructs could be directly used for infectivity assays through carborundum-mediated mechanical inoculation on *Chenopodium quinoa* and *Nicotiana benthamiana* and also the subsequent sequencing. After verification of the infectivity of the selected clones, the full-length sequences of each clone were determined by the Sanger sequencing technique as described previously [25]. For convenience in the subsequent inoculation assays, the full-length genomes of BaMV Vietnam isolates in the infectious pCass-based constructs were sub-cloned into the *Hin*dIII-*Xba*I site of pKn vector, which allows for *Agrobacterium*-mediated inoculation on *N. benthamiana*, as described previously [27]. Briefly, cultures of *A. tumefaciens* cells harboring the pKn-based infectious constructs of BaMV Vietnam isolates were adjusted to a concentration with the optical density at 600 nm of 0.1 (OD_600_ = 0.1) and infiltrated into the underside of *N. benthamiana* leaves by using a 1 mL syringe without needle. The inoculated plants were cultivated in a growth room maintained at 25–28 °C with 16/8 h light/dark cycles.

### 2.3. Sequence Analysis

The nucleotide sequences of BaMV Vietnam isolates and satBaMV RNAs were analyzed using EMBOSS package version 6.5 [28], BioEdit version 7.2.5 [29], and MEGA X version 10.0.5 [30]. For pair-wise alignments, the tool Needle in the EMBOSS package was used with default values for gap opening and extension penalties (10 and 0.5, respectively). For multiple sequence alignments, the program CLUSTAL W [31] provided in the MEGA X package was used with default parameters, with gap opening/extension penalties of 10/0.1 and 10/0.2 for pair-wise and multiple alignment stages, respectively. The results were illustrated using the software GeneDoc version 2.7 [32]. Phylogenetic tree reconstructions were performed using the neighbor-joining and maximum likelihood algorithm provided in the MEGA X package.

### 2.4. Data Visualization

To analyze the main variable regions in the BaMV genome that affect the geographical classification of BaMV, different regions of the Vietnam isolate of BaMV (BaMV-VN2-9) were used as the query sequence to compare with the corresponding regions of other known BaMV strains/isolates. The results of sequence comparisons were visualized by the heat map generated by the heatmap.2 function natively provided in R [33], with column and row dendrograms to depict the effect of different regions and clustering of BaMV strains/isolates, respectively.

### 2.5. Identification of Main Variations in BaMV Genome

To identify the main regions with differences in genomic sequences between the Vietnam isolates and other known BaMV strains, the consensus sequences for BaMV strains/isolates from different geographical or host origins were generated using the software BioEdit [29] and aligned using CLUSTAL W [31]. The highly variable region was presented using the software GeneDoc [32]. For the analysis of nucleotide polymorphism and divergence, the software DnaSP 6.0 [34] was used. Selection pressure imposed on different BaMV ORFs was analyzed using the synonymous/nonsynonymous substitution analysis in DnaSP 6.0 [34] and the codon-based Z-test for selection in MEGA X [30] software using default parameters.

### 2.6. Inference of Phylogenetic Relationships

Phylogenetic analyses on the full-length or partial genomic sequences of BaMV strains/isolates were performed using the software MEGA X [30]. The phylograms were generated using the maximum likelihood algorithm, and the resulting topologies were verified with neighbor-joining and maximum parsimony algorithms.

## 3. Results

### 3.1. Construction of Infectious Clones of BaMV Vietnam Isolates

Bamboo plants (*B. funghomii*) showing severe mosaic symptoms suspected of BaMV infections were observed in Xuan Cam Commune, Hiep Hoa District, Vietnam (Figure 1a). Leaf samples were collected from two independent bamboo bushes, designated as VN1 and VN2. The presence of BaMV in these samples was examined by Western blot analysis for the expression of BaMV CP as described previously [35] and verified by mechanical inoculation of the leaf saps onto *C. quinoa* and *N. benthamiana*. As shown in Figure 1b, protein bands with similar electrophoretic mobility to BaMV CP were detected in both the bamboo leaf samples by specific antiserum, suggesting the existence of BaMV in the bamboo leaf samples from Vietnam. Total RNAs were extracted from the bamboo leaf samples and subjected to cDNA synthesis and 5′-RACE using the BaMV consensus primer (Ba652R). New primers were designed based on the 5′-terminal sequences of the Vietnam isolates of BaMV, and the full-length cDNAs of the Vietnam isolates of BaMV, designated BaMV-VN1 and BaMV-VN2 were synthesized and cloned into pCass vector [26] to generate infectious clones. Six independent clones for BaMV-VN1 (BaMV-VN1-1 to -6) or BaMV-VN2 (BaMV-VN2-1 to -3 and BaMV-VN2-7 to -9) were randomly selected for inoculation assays and sequencing analysis. Subsequently, the full-length BaMV genomes on the pCass-based constructs were sub-cloned into the pKn vector [27], which allows for *Agrobacterium*-mediated inoculation, as described above, for the convenience of preparing inoculum. The results of inoculation assays using the pKn-based infectious clones on *N. benthamiana* showed that all the clones of Vietnam isolates of BaMV elicited similar, but slightly milder, symptoms on the test plants as those produced by the BaMV-S [36], the type strain of BaMV, suggesting the genome integrity of these clones. The representative result of the inoculation assay with BaMV-S and BaMV-VN2-9 is shown in Figure 1c,d, respectively.

### 3.2. Sequence Analyses and Phylogeny Reconstruction of the Vietnam Isolates of BaMV

Following the verification of genome infectivity by inoculation assay, the full-length sequences of these infectious clones were obtained with the Sanger sequencing technique [37]. To account for the within-sample variations, all the six infectious clones for each sample (BaMV-VN1 or BaMV-VN2) were sequenced. Pair-wise sequence comparison revealed that clones within the same sample source shared over 99% sequence identities, with sporadic point-mutations throughout the genome. The clones from BaMV-VN1 share 95–96% sequence identities with those from BaMV-VN2. The Vietnam isolates of BaMV are very distinct from the BaMV strains/isolates from other regions in the world, sharing only 76–78%, 75–78%, and 77–80% of full-length, ORF1, and CP sequence identities, respectively, indicating that the Vietnam isolates of BaMV may have an evolutionary history different from those of other known BaMV strains/isolates. To explore the phylogenetic relationships between the Vietnam isolates and other BaMV strains/isolates, the full-length genomic sequences available in GenBank (Appendix A) were aligned by CLUSTAL W [31], with three other potexviruses closely related to BaMV [10], namely plantago asiatica mosaic virus (PlAMV, LC422371), alternanthera mosaic virus (AltMV, MH423501), and foxtail mosaic virus (FoMV, NC_001483) as the outgroup, and the phylogenetic tree was reconstructed using the maximum likelihood method and verified by the neighbor-joining method in the MEGA X package [30]. The result revealed the clustering of the BaMV Vietnam isolates into BaMV-VN1 and BaMV-VN2 clades, which form the sister clade to the BaMV-Yoshi isolates (BaMV-MT clade) found in California, USA (Figure 2). As shown in the phylogenetic tree, the Vietnam and Yoshi isolates of BaMV are distantly related to the other known BaMV strains/isolates. Based on the species demarcation criteria of the genus *Potexvirus* [38], the viruses in two different species should share less than 72% sequence identities in the coat protein or ORF1 protein genes. Thus, the Vietnam isolates are still classified within the species *Bamboo mosaic virus* and not new species of the genus *Potexvirus*.

### 3.3. Detection of Unique Satellite RNAs Associated with BaMV-VN1 and BaMV-VN2

To test whether satRNAs were also associated with BaMV-VN1 and BaMV-VN2, 5′-RACE with BaMV satRNA consensus primer (BS-40: BS40: 5′-TCAACTGGTTGGTG-CACGGT-3′) was performed using total RNAs extracted from the original bamboo leaf samples as the templates, and the resulting amplicons with expected length of 399 base-pairs were sequenced as described above. The result confirmed that satllite RNAs, designated satBaMV-VN1 and SatBaMV-VN2, were indeed associated with BaMV-VN1 and -VN2, respectively. To analyze the full-length satRNA sequences, specific 5′ primers, (5′-GCAGGCCTGGAAAACCAACAGAA-ACG-3′and 5′-GCAGGCCTGGAAAACCAACGGAACAAAACG-3′, were designed for satBaMV-VN1 and satBaMV-VN2, respectively, based on the 5′-RACE result and used in RT-PCR with oligo d(T)_18_ to amplify the full-length cDNAs of satBaMV RNAs. The resulting amplicons were cloned into the pCass vector and sequenced as described above. Six colonies each were randomly picked from plates containing *Escherichia coli* harboring constructs of satBaMV-VN1 or satBaMV-VN2 and subjected to nucleotide sequencing and phylogenetic analysis. The result revealed that the Vietnam isolates of satBaMV RNAs associated with the same BaMV (either BaMV-VN1 or BaMV-VN2) share over 98% sequence identities with each other, but only about 91% identities with those associated with different BaMV Vietnam isolates. Furthermore, the Vietnam satBaMV RNAs are distinct from the satBaMV RNAs reported previously [9,10], clustering into a unique clade (Figure 3, clade IV) in the phylogenetic analysis, similar to those observed for BaMV genomic RNAs (Figure 2). The closest relative to the satBaMV RNAs associated with BaMV-VN1 is the satBaMV-MAZSL1 (Accession number: KU870665, 91% sequence identity) found in Ma bamboo (*Dendrocalamus latiflorus*) from China [7], whereas those associated with BaMV-VN2 share the highest sequence identities with satBaMV-Y20 (MT111574, 91% sequence identity) associated with BaMV-Yoshi found in California, the United States, in accordance with its helper virus BaMV-VN2. The findings suggest that the satBaMV RNAs currently identified in Vietnam may have diverged from the other satBaMV RNAs either at an early stage of their evolutionary history or with much higher mutation rates compared to other satBaMV RNAs. These observations also confirm that satBaMV RNAs are widely associated with BaMV from different geographical origins.

### 3.4. Identification of a Highly Variable Region in BaMV Genome

Several phylogenetic analyses have been performed on BaMV strains/isolates collected from different regions of the world [5,6,9,10,11]. However, the mutational hotspots in the BaMV genome that may demarcate the clustering of different BaMV isolates remain to be explored. It has been shown that mutational hotspots in the genome, characterized by the occurrence of mutations at the same region of the genomes of different lineages of a species, may reflect the underlying driving force of evolution, which can make evolution very repeatable [40]. These mutational hotspots may also be observed as the highly variable region(s) in the genomes of different lineages in a species. To identify such regions, different ORFs or fragments on the genome of BaMV-VN2-9 were used as the queries to compare with the corresponding regions of other BaMV strains/isolates. The genomic sequences of three potexviruses are closely related to BaMV [10], PlAMV (LC422371), AltMV (MH423501), and FoMV (NC_001483) were also included as outgroup controls.

The results of sequence comparisons are summarized in Appendix A. To visually represent the effects of each region on the grouping of BaMV strains/isolates in phylogenetic analysis, a heatmap was generated (Figure 4) using the percent identity values in Appendix A. As shown in the hierarchical clustering of different regions (Figure 4, the dendrogram on the top), the ORF1 coding region exhibited the most similar pattern as that of the full-length genome and is thus clustered within the same clade with a full-length genome. In contrast, the 3′-UTR is highly conserved among BaMV strains/isolates and thus contributed little to the grouping of the BaMV strains/isolates. On the other hand, the TGBp3 coding region is the least conserved among the BaMV strains/isolates; however, the similarity pattern generated using the TGBp3 coding region does not resemble that generated using full-length sequences. Likewise, the similarity patterns generated by regions other than ORF1 coding do not reflect well that generated by full-length sequences. Thus, the observations suggested that ORF1 coding sequences may contain certain mutational hotspots as the key determinant for the classification of BaMV into different clades.

To identify the mutational hotspots in the BaMV genome, the nucleotide polymorphism and divergence were analyzed using the program DnaSP 6.0 [34], with a sliding window length and step size of 100 and 25 nucleotides, respectively, using default parameters. The result of average nucleotide divergence analysis over the genome of all BaMV isolates revealed the presence of a highly polymorphic region near nucleotide positions (nts) 1200–1600 (Figure 5a) relative to the multiple sequence alignment. Further analysis of overall nucleotide divergence between BaMV-VN2 and other different subgroups (as shown in Figure 2) of BaMV isolates showed that the most divergent region between BaMV-VN2 isolates and other non-Vietnam subgroups of BaMV resides also approximately at nts 1200–1600 region (Figure 5b). However, the most divergent region between BaMV-VN2 and BaMV-VN1 is located in the CP coding region (Figure 5b, light green line), suggesting the influence of different evolutionary pressures on Vietnam isolates as compared to other BaMV isolates.

To further validate the candidate key region that may determine the clustering of BaMV isolates, consensus sequences were generated for each clade as shown in Figure 2 and subjected to multiple sequence alignments. As shown in Figure 6, the region between nts 1300 to 1720, relative to the numbering of the multiple-sequence alignment, was found to be highly variable, which is in close proximity to the highly polymorphic region identified in the nucleotide diversity analysis (nts 1200 to 1600, Figure 5a). There appeared to be large fragments of deletion (nts 1530 to 1562) and insertion (nts 1653 to 1682) in the Vietnam isolates of BaMV (Figure 6a); however, the alignment of amino acid sequences of ORF1 of different BaMV did not show large gaps (Figure 6b), suggesting that the length of the region within ORF1 is important so that the deletion has to be compensated by the insertion nearby to maintain the length of the coding region. For comparison, phylogenetic analyses were performed with MEGA X using different ORFs of BaMV. The resulting phylograms generated by the neighbor-joining method are shown in the Appendix A. As shown, the clustering of BaMV isolates in the phylogenetic tree for ORF1 (Appendix A) resembles that for the full-length genome (Figure 2), whereas those generated with nucleotide sequences of other ORFs contain different clustering patterns (Appendix A, as indicated by the red arrows on the right) as compared to that for the full-length genome (Figure 2).

To analyze whether this highly variable region was under different selection pressure compared to the other ORFs of BaMV, synonymous/nonsynonymous substitution analysis was performed using both the DnaSP 6 [34] and MEGA X [30] software. The result revealed that all ORFs of BaMV are under selection pressure, with Ka/Ks (also known as dN/dS) ratios [41] significantly lower than 0.3 (*p*-value < 0.0001) for all possible pairs of BaMV isolates. In addition, although the Ka/Ks ratios for the highly variable region are higher, ranging from 0.1 to 0.86, they are still significantly lower than 1 (*p*-value < 0.0001), indicating that the mutations in all ORFs of BaMV, including the highly variable region, are under strict constraint to maintain the amino acid encoded (i.e., synonymous substitutions).

To test whether this highly variable region may reflect the evolutionary history of BaMV isolates, phylogenetic trees were generated based on this region and compared with that reconstructed using the full-length sequences. To simplify the tree topology for comparison, consensus sequences were generated for each main clade, as indicated in Figure 2. BaMV-TW contains BaMV strains/isolates reported in Taiwan and Indonesia; BaMV-CN represents those found in Fujian Province, China, excluding those within the BaMV-KU and BaMV-KX clades, which are isolates from Ma bamboo (*Dendrocalamus latiflorus*) and the recently identified BaMV isolates in Fujian, respectively; BaMV-MT includes those identified in California, USA; and BaMV-VN1 and -VN2 include the Vietnam isolates from the two samples identified in this study. Following the multiple-sequence alignments of the consensus sequences of the full-length genome or the highly variable regions, phylogenetic trees were reconstructed with the Maximum Likelihood method in MEGA X [30]. As shown in Figure 7, the phylogenetic tree generated using the full-length sequences (Figure 7a) was closely reflected by the one generated using only the highly variable regions (Figure 7b). The result suggested that the highly variable region may represent the main events in the divergence and evolutionary history of BaMV isolates from different host species and geographical distributions.

## 4. Discussion

BaMV is the major viral pathogen of bamboos and severely affects the quality and yields of bamboo products [42,43]. As the bamboo industries continue to prosper, there is a growing concern that BaMV may also be transmitted through raw products. It is important to keep an inventory of the occurrence of BaMV around the world. In this study, we have identified unique isolates of BaMV and the associated satBaMV RNAs from an additional bamboo species, *B. funghomii*, in Vietnam, thus expanding the known host range and geographical distribution. Sequence analysis showed that the Vietnam isolates are distantly related to the other known BaMV isolates or satBaMV RNAs, forming a distinct clade in the respective phylogenetic analysis. The analysis also revealed a highly variable region in the ORF1 coding region, about 420 nts in length, which may affect the classification of BaMV isolates.

There is a close resemblance between the phylogenetic trees constructed using the consensus full-genome sequences and the highly variable regions in the ORF1 coding sequence of BaMV isolates from different geographical origins (Figure 7). This observation suggested that this highly variable region is strongly affected by the evolutionary pressure and that the sequence variations may have main contributions to the fitness of BaMV in a specific environment. However, the results of 3D structural modeling analyses using the Phyre2 tool [44] did not identify acceptable structure templates in this region, suggesting that this highly variable region might not share significant structural similarity to any known proteins.

Based on the reported organization of functional domains of ORF1 protein [12], this highly variable region, amino acid residues 440–520, corresponds to the hydrophilic region in between the methyl transferase (capping enzyme) domain and the helicase-like domain. Thus, this region may exhibit high structural flexibility, serving as the hinge that connects two functional domains and thus allowing high sequence variability. The analysis of selection pressure revealed that all ORFs of BaMV, including the highly variable region, are under purifying selection pressure. This is consistent with the report that purifying selection pressure is most prevalent in nature [45,46]. As shown in a recent study, many human RNA viruses are under strong purifying selective pressure [47]. Here, we provide evidence supporting that BaMV is also under purifying selection pressure, despite the presence of a highly variable region in the genome. The highly divergent (Figure 5 and Figure 6) but synonymous substitutions mutations suggested the functional requirement of the ORF1 protein, which is the major component of BaMV replicase. The reason for the high degree of divergence in nucleotide sequence but conservation of amino acid sequence may reflect the interaction with the host or environmental factors for better adaptation. It has been shown that the synonymous mutations might exert their effects on the fitness of an organism through several mechanisms [48], including the modification of the mRNA structure/stability, alteration of translation efficiency, interactions with small RNAs (including microRNAs or small interfering RNAs, abbreviated as siRNAs), and change in substrate specificity. For example, different haplotypes of human catechol-O-methyltransferase may alter mRNA secondary structure and thus modulate protein expression efficiency [49], and a synonymous single nucleotide polymorphism in the multidrug resistance 1 gene may affect the timing of co-translational folding of the mRNA and insertion of the protein product into the membrane, thereby altering the substrate specificity [50]. Similarly, the synonymous substitutions in the highly variable region in the BaMV genome may hint at the interactions between BaMV and the host plants. Different bamboo species may encode different isoaccepting transfer RNAs (tRNAs), which possess different anticodons but transfer the same amino acids, for the efficient translation of important viral proteins, or the bamboo plants may express a subset of tRNAs with different abundances under the influence of different environmental factors related to geographical distributions. However, only the hinge region exhibits a high level of nucleotide diversity, suggesting that other factors may affect the fitness of such synonymous mutations.

The other important candidate driving force for the synonymous substitutions is the siRNA-based host defense system. A comprehensive analysis of the BaMV-derived siRNAs in *N. benthamiana* and *Arabidopsis thaliana* [51] has demonstrated that most of the BaMV-related siRNAs in *A. thaliana* were derived from the 5′-half of the BaMV genome, with the highest amount of both (+)- and (−)-strand siRNA detected around the region corresponding to the highly variable region identified in this study (approximately nts 1200–1720) (Figure 5 and Figure 6). In contrast, BaMV siRNAs in *N. benthamiana* were mainly derived from the 3′-terminal region (CP and 3′-UTR). The observations indicated that the RNA silencing systems in different plants, including different bamboo species, may exhibit preferences against different genomic regions of the invading virus. The reason for the presence of the mutational hotspot in the ORF1 coding region might be that the genomic RNA, which serves as the mRNA for ORF1 protein, is more vulnerable to the attack of miRNAs or siRNAs of the host defense system, as compared to the viral subgenomic RNAs that are highly efficiently transcribed from the internal sub-genomic promoters on the minus-strand complementary to the genomic RNA. Thus, the degradation or sequestration of viral genomic RNAs might pose a greater threat to the survival and fitness of BaMV or related potexviruses. In response, BaMV and the related potexviruses might have evolved the high variability in the ORF1 coding sequence to evade the miRNA- or siRNA-mediated defense system of the hosts. The observation that the mutational hotspot in Vietnam isolates of BaMV resides in the CP coding sequence (Figure 5b, light green line) may indicate the presence of different preferences of siRNA-based defense systems in the different bamboo species, *B. funghomii*. The result of the synonymous/nonsynonymous substitution analysis using the CP coding sequences of BaMV Vietnam isolates showed that the Ka/Ks ratio ranged from 0 to 0.37 for all possible pairs, significantly lower than 1 (*p*-value < 0.0001), suggesting that this region, similar to the coding regions of other BaMV proteins, is also under purifying selection pressure to maintain the amino acid sequence with variations of the nucleotide sequence. This observation provided further support for the hypothesis that the miRNA- or siRNA-based defense systems in different hosts may play a role in the adaptation of BaMV. However, other possibilities, such as the interactions with other host or insect factors, could not be ruled out.

It is worth noting that the two BaMV strains in the BaMV-KU group were identified from a different bamboo host, Ma bamboo (*D. latiflorus*) [7], within the same Fujian Province of China as those from other bamboos [6], indicating that Ma bamboo may have posed a different selection pressure on the variation of BaMV. On the other hand, the BaMV strains within the BaMV-CN, -KX, -KT groups isolated from the Fujian Province of China were from bamboo hosts belonging to different genera [7]. For example, the three strains in the BaMV-KT group were collected from *Bambusa rutile*, *Phyllostachys aureosulcata*, and *Bambusa rigida*; however, the genomic sequences of the three BaMV strains are highly similar to each other (90–94.7% identities) and clustered within the same BaMV-CN group with KX648528, KX648529, and KX648531 collected from *Dendrocalamus tsiangii*, *Neosinocalamus affinis*, and *Bambusa xiashanensis*, respectively. This observation suggested that both geographical and host factors may have contributed to the diversification of BaMV.

## 5. Conclusions

In conclusion, unique Vietnam isolates of BaMV, together with the associated satellite RNAs, infecting *Bambusa funghomii* were identified in this study for the first time to our knowledge. The finding further supported the notion that BaMV is naturally present in the main plantations of bamboos worldwide and that both geographical and host factors may contribute to the diversification of BaMV. A highly variable region corresponding to the hinge between the methyl-transferase and helicase motifs was identified that might reflect the interactions between BaMV and host plants and/or environment in the evolutionary history. Altogether, this study provides further information for the comprehension of the distribution and diversification of BaMV.

## Figures and Tables

**Figure 1 viruses-14-00698-f001:**
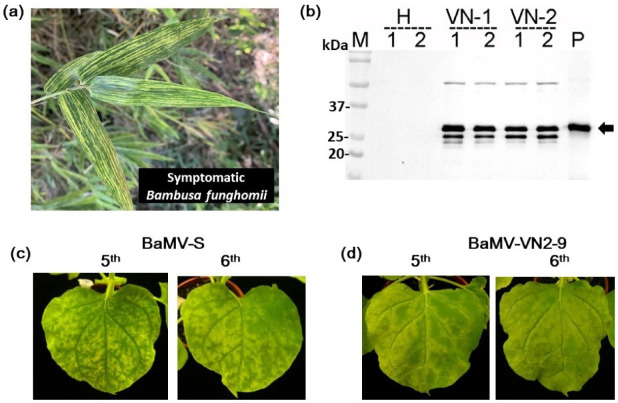
Symptoms on *Bambusa funghomii* and *Nicotiana benthamiana* caused by *Bamboo mosaic virus* (BaMV) Vietnam isolates. (**a**). Leaves of *B. funghomii* with severe mosaic symptoms in the bamboo plantations of Xuan Cam Commune, Hiep Hoa District, Vietnam, were presented. (**b**). Western blot analysis of the samples using specific antiserum against BaMV coat protein (CP). M, size marker; H, protein sample extracted from healthy *N. benthamiana*; VN-1 and VN-2, samples from the two independent bamboo bushes, with 2 replicates; P, BaMV virion (50 ng) as the positive control. The expected position of BaMV CP is indicated by the arrow on the right. (**c**,**d**). Inoculation assays on *N. benthamiana*. The pKn-based [27] infectious clones of BaMV Vietnam isolates (six independent clones each for BaMV-VN1 or BaMV-VN2) were constructed and subjected to inoculation assay on *N. benthamiana* through *Agrobacterium*-mediated infection. All infectious clones of BaMV Vietnam isolates inflicted similar symptoms milder than that caused by BaMV-S, the type strain of BaMV. The symptoms caused by BaMV-VN2-9 are shown as an example. Top view of the symptoms on whole plants (**b**) and close-up views of the symptoms on the 5th and 6th leaves (**c**,**d**) are shown. The identities of the infectious clones are indicated on the top.

**Figure 2 viruses-14-00698-f002:**
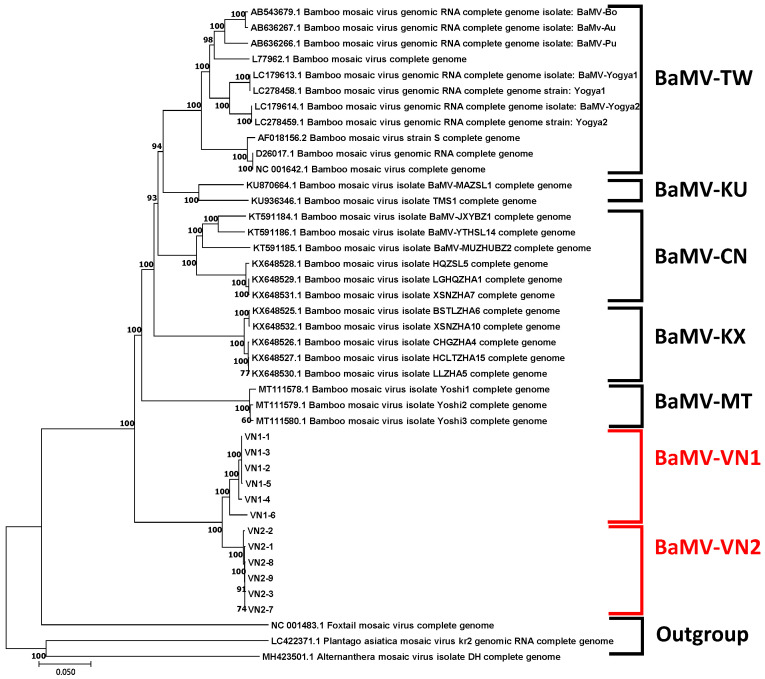
Molecular Phylogenetic analysis of BaMV using the full-length genomic sequences by Maximum Likelihood method. The full-length genome sequences of BaMV strains/isolates, as listed in Appendix A, were aligned using CLUSTAL W [31] and subjected to phylogenetic relationship inference using the Maximum Likelihood method based on the Tamura-Nei model [39] in the MEGA X software [30]. The horizontal branch lengths are drawn to scale, relative to the scale bar at the bottom, representing 0.05 substitutions per site. The major clades formed by BaMV strains/isolates are indicated on the right, as follows: BaMV-TW, strains/isolates reported in Taiwan and Indonesia; BaMV-CN, strains/isolates from Fujian Province, China, excluding those within the BaMV-KU and BaMV-KX clades; BaMV-KU, strains from Ma bamboo (*Dendrocalamus latiflorus*); BaMV-KX, isolates recently identified in Fujian [6]; BaMV-MT, isolates identified in California, USA; and BaMV-VN1 and -VN2, Vietnam isolates from the two samples identified in this study. Three potexviruses closely related to BaMV were used as the outgroup in the phylogenetic analysis.

**Figure 3 viruses-14-00698-f003:**
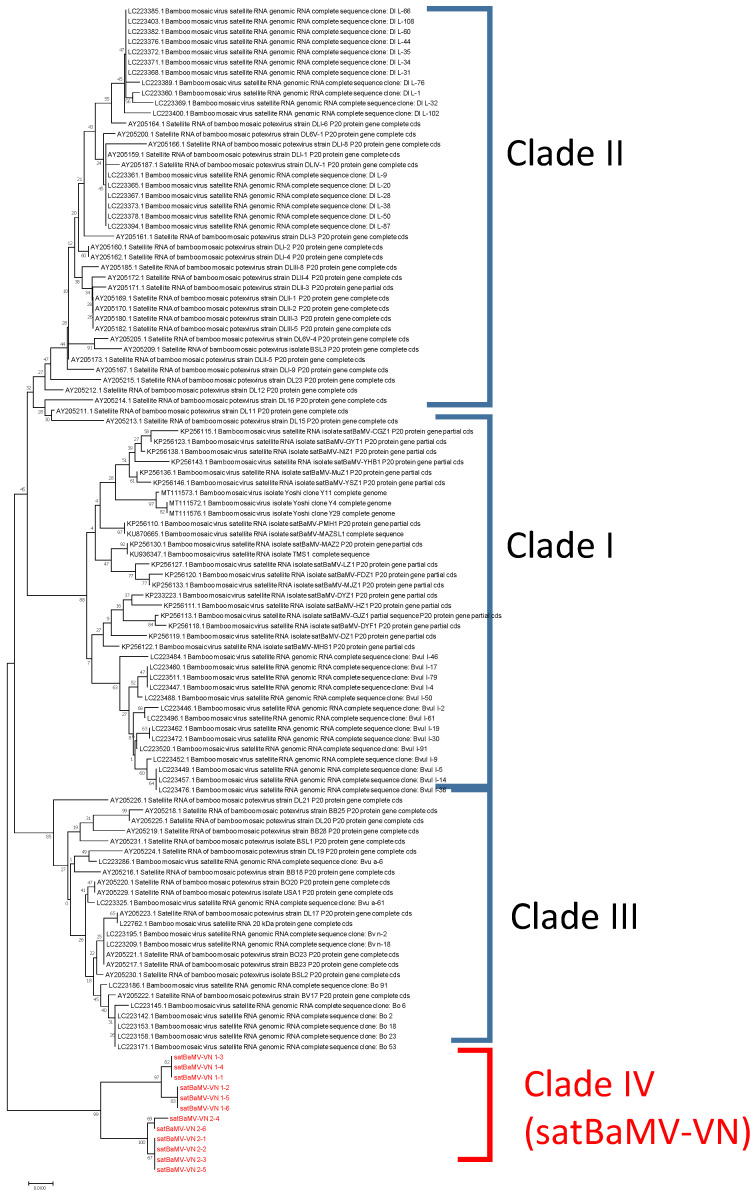
Phylogenetic analysis of satellite RNAs associated with BaMV (satBaMV RNAs). Nucleotide sequences of representative satBaMV RNAs from previous reports [9,10] and those associated with BaMV-VN1 or BaMV-VN2 were subjected to phylogenetic analysis using MEGA X [30], and the phylogenetic tree generated with the neighbor-joining algorithm is shown. The three main clades reported previously plus the clade consisting of the satBaMV RNAs associated with BaMV Vietnam isolates are indicated on the right. The horizontal branches, representing the genetic distances, are drawn to scale. The scale bar at the lower left represents 0.01 substitutions per site. The topology of the tree was evaluated with 100 bootstrap replicates. The percentage of replicate trees that maintain the same clustering in the bootstrap test is indicated on the respective nodes.

**Figure 4 viruses-14-00698-f004:**
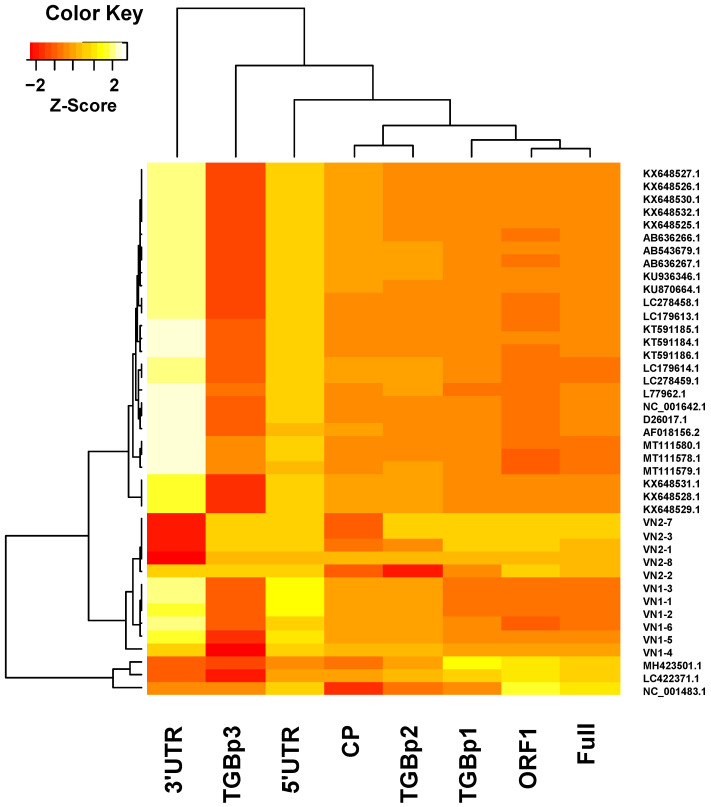
Heatmap representation of the similarities between different regions of BaMV-VN2-9 and the corresponding regions of other BaMV strains/isolates or closely related potexviruses. To identify the region in the BaMV genome that closely represents the divergence of BaMV genomes, different regions (as indicated at the bottom) of BaMV-VN2-9 were compared with the respective regions in other BaMV strains/isolates or closely related potexviruses. The sequence similarities were normalized to different regions of the BaMV genome as z-scores [33] and shown in the heatmap, with different colors as indicated on the top-left. The dendrograms on the top and left of the heatmap illustrate the hierarchical clustering based on the different regions or BaMV strains/isolates, respectively.

**Figure 5 viruses-14-00698-f005:**
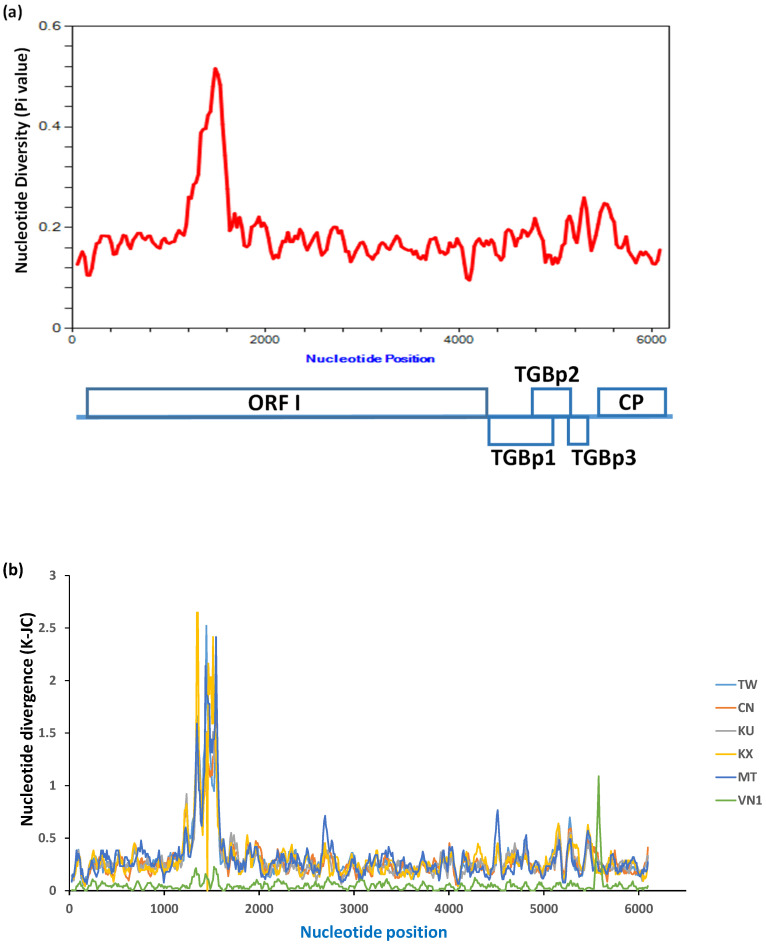
Analysis of nucleotide divergence of BaMV genome sequences. (**a**). Overall nucleotide diversity of genomic sequences of BaMV isolates. The average number of nucleotide differences per site between all possible pairs (Pi value) [41] of BaMV genomic sequences is plotted in the line chart. A sliding window of 100-nt in size and 25-nt in step was used to calculate the Pi value over the genomic sequences of 38 BaMV isolates (excluding BaMV-Au, GenBank accession number AB636267, which harbors an abnormally short ORF1 gene of only 1719 nucleotides in length that possibly resulted from artifacts in the sequencing process). A schematic representation of the BaMV genome depicting the approximate positions of each ORF is shown below. (**b**). Overall nucleotide divergence between BaMV-VN2 group and other BaMV groups. The nucleotide divergence with Jukes and Cantor correction (K-JC) values [41] between BaMV isolates in the VN2 group and those of other BaMV groups were calculated and plotted. A sliding window of 100-nt in size and 25-nt in step was used for the calculation. The intergroup K-JC values between various BaMV groups and BaMV-VN2 group are represented in different colors as indicated on the right. The relative nucleotide positions are indicated on the *x*-axis.

**Figure 6 viruses-14-00698-f006:**
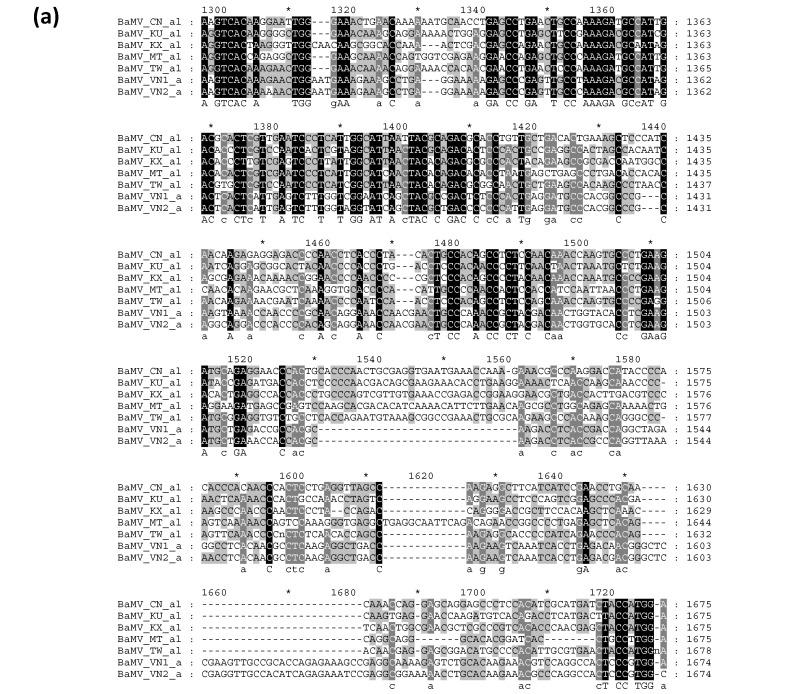
Multiple sequence alignment depicting the highly variable regions in the open reading frame 1 (ORF1) of the Bamboo mosaic virus (BaMV) genome. The consensus sequences of nucleotides (**a**) and amino acids (**b**) of BaMV isolates from different geographical origins were aligned using the software Custal W [31] and illustrated using the software GeneDoc [32]. Identical and consensus residues are shown in black and grey backgrounds, respectively. Deletions were represented by the “-” sign. The names and the relative nucleotide positions are shown on the left and right of each sequence, respectively. The sequence at the bottom of each block represents identical (upper case) and consensus (lower case) residues.

**Figure 7 viruses-14-00698-f007:**
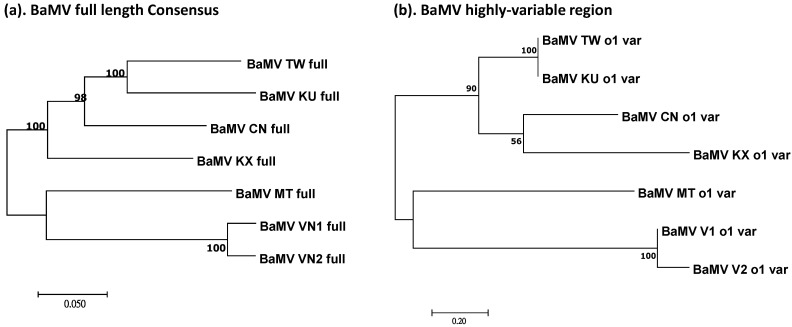
Molecular phylogenetic analysis of BaMV using consensus full-length sequences or the highly variable regions of the ORF1 coding sequence by Maximum Likelihood method. The phylogenetic relationship of BaMV strains was inferred from the full-length sequences (**a**) or the highly variable regions of ORF1 (**b**). For clarity, consensus sequences of BaMV strains/isolates were generated based on the clustering of the full-length BaMV genomes. These consensus sequences were then aligned using CLUSTAL W [31] and subjected to infer phylogenetic relationship using the Maximum Likelihood method based on the Tamura-Nei model [39]. The horizontal branch lengths are drawn to scale, relative to the scale bar at the bottom, representing the number of substitutions per site. Evolutionary analyses were conducted in MEGA X [30]. The suffixes “full” or “o1 var” to the name of the sequence indicate the phylogenetic relationship reconstructed using the full-length or the highly variable region of ORF1 coding sequences, respectively.

## Data Availability

All nucleotide sequences of the Vietnam isolates of BaMV and satBaMV RNAs have been deposited in GenBank, under the following accession numbers: MZ463301 to MZ463306 for BaMV-VN1-1 to 6; MZ463307-MZ463309 for BaMV-VN2-1 to 3; MZ463310 to MZ463312 for BaMV-VN2-7 to 9; MZ463313 to MZ463318 for satBaMV-VN1-1 to 6; and MZ463319 to MZ463324 for satBaMV-VN2-1 to 6.

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
