# Peer review of "First Report of Distinct *Bamboo mosaic virus* (BaMV) Isolates Infecting *Bambusa funghomii* in Vietnam and the Identification of a Highly Variable Region in the BaMV Genome"

_viruses, 2022, doi:10.3390/v14040698_

Round 1

Reviewer 1 Report

The revised manuscript by Huang and colleagues now shows a greater consistency with the title and the aim of the study. The analyses were largely improved and are clearly presented.

I only have minor modifications to propose.

The description of the material used in this study should be improved.

Were really 2 samples collected from different (?) symptomatic bamboo bushes (line 79)  or were 2 leaf samples positive for BaMV detection among different samples collected?

It is still not clear what the infectious clones are made of? From the "materials and methods" section it is not known, from the result section (line 147) one can find out that they are cDNAs under the control of the 35S promoter and terminator, in a classical plasmid (pKS / pUC19 kind) and from legend of fig 1 one conclude that they are agro-infiltrable infectious clones that is cDNA in a binary vector (containing T-DNA with left and right borders). Were really C. quinoa plants agro-infiltrated? Please, explain clearly the cloning of the infectious clones and the inoculation procedure in the materials and methods section.

Number supplemental figures (or is it a single figure? iI this is the case name panels) and quote figure numbers/panel names in the text.

Fig2: Align brackets carefully with the isolate names.

Fig3: The resolution of the figure didn't allow me to check whether it corresponds to the statements given in the text.

Fig.5 Could be easier to understand with a genetic map aligned to the nt positions to help readers locate the coding regions. This could be added between the two panels and aligned to both graphs. AB636266 (which should be called BaMV-Pu) was excluded, it is not clear why.

Lines 451-453: In their discussion, the authors link the mutational hotspot of the Vietnam isolates in the CP coding region to differences in RNAi systems in different bamboos. However, nothing is said about the diversity of the CP at the amino-acid level. If the proteins are different between the Vietnam isolates and other isolates worldwide, this could reflect a selection pressure due to different interactants, either at the plant level (e.g. for cell-to-cell or long-distance movement) or at the level of the dipteran insects suggested to transmit the virus. Also, if divergence is at the nt level, alternative scenarios are possible like for the ORF1 coding region; translation, interaction with host factors involved in replication…  

Additional comments:

Lines 48, 63, 85, 118, 139, 147, 186, 226, 298, 303, 356, 405, 410, 434, 437, 444: wrong hyphenations

Line 53: for long-distance movement of what? the BaMV virus or the satRNA?

Lines 55 & 56: are? remained?

Lines 74 & 75: reflects? provide?

Lines 95-96: what are the infectious clones made of? is the viral cDNA under the control of an in vitro transcription promoter like T7 promoter under the control of a 35S in vivo promoter? or is it an agro-infiltrable infectious clone? This should be mentioned. Also, the way of plant inoculation should appear as a short sentence in the materials & methods section.

Line 111: the Vietnam isolate is defined => replace "a" with "the".

Line 125: correct codon-based.

Line 134: check the police of Vietnam isolates.

Lines 159-161: Add the meaning of H samples.

Line 147: the type and name of the plasmid vector used for obtaining infectious clones should appear in the mat & meth section.

Line 149: remove "each".

Lines 154, 165, 166: replace "was" with "is" or "are".

Lines 164, 171: a space is missing.

Line 169: change integrity with infectivity.

Line 175: remove "however" as there is no contradiction/opposition with the previous statement.

Line 190: "However" is probably not appropriate here as this is not contradictory.

Lines 193-194: species and genus names should be italicized.

Lines 209-210: specify on what material the 5'RACE was performed. What is the length of the amplicons analyzed here?

Line 229: correct share.

Line 231: replace suggested with suggest.

Line 232: correct redundancy.

Line 235: replace demonstrated with demonstrate. Replace with confirm?

Line 256: correct were.

Line 260: correct are.

Lines 278-279: In fig 5a it rather seems to be 1200-1600.

Line 293: could it be that some regions are just swapped with no specific length selection in ORF1? A shorter overall genome is more rapidly replicated and this could be sufficient to become predominant. This applies to any part of the genome of any virus.

Line 297: replace were shown with are shown. The same throughout the manuscript and fig legends.

Line 351: correct represents.

Line 447: replace "minus-strand of genomic RNA" with "minus-strand complementary to the genomic RNA".

Line 473: add was before identified.

Line 475: change together with altogether and provided with provides.

Reviewer 2 Report

This MS is a revised version of a previously reviewed MS. In this revised MS, all problems I pointed out in the review were well addressed. The new results and their discussions in the revised MS are fascinating. I strongly recommend publishing the manuscript in present form.

Author Response

Our response:

We sincerely thank Reviewer 2 for providing helpful instructions to improve the quality of this manuscript. The comments of Reviewer 2 are greatly appreciated.

Sincerely yours,

Hu, Chung-Chi

Professor

Graduate Institute of Biotechnology,

National Chung Hsing University,

Taichung, 40227                                         

Taiwan

This manuscript is a resubmission of an earlier submission. The following is a list of the peer review reports and author responses from that submission.

Round 1

Reviewer 1 Report

In their manuscript entitled "First report of distinct bamboo mosaic virus (BaMV) isolates from Vietnam and the identification of a genetic marker for the classification of BaMV with different origins" Huang et al. identified new isolates of BaMV in Vietnam where the virus had never been officially documented before. They collected samples from 2 bamboo plants in the fields of Xuan Cam Commune, cloned the cDNAs to obtain both sequence data and infectious clones and characterized the isolates. They showed that the Vietnam isolates are quite different from other known isolates identified in other countries, that they are associated with satellite RNAs and that one isolate probably arose from a recombination event between a BaMV and an AltMV. They also identified a region in ORF1 that is variable among isolates from different clades and well reflects the overall variability of known isolates.

The authors present their work as aimed at analyzing the occurrence of BaMV in Vietnam and establishing genetic markers of the geographical origin of different isolates. They stress the importance, if BaMV is present in Vietnam, of establishing the phylogenetic relationships between Vietnam isolates and those from other geographical areas in the world to get insight into the genetic variabilities and molecular evolution of BaMV (lines 61-63).

However only two bamboo leaf samples were collected from a single field to start this study. It is not clear whether many plants were monitored and collected from all over Vietnam but only two were positive or if only two were collected from that specific area, in a targeted nonsystematic approach. Thus, how much are these two plants representative of the occurrence of BaMV in the country or region? And how robust the genetic markers will then be to discriminate Vietnamese origin of the virus? How representative of all Vietnam BaMV isolates is the variability the authors describe in this paper? Although the BaMV isolates are interesting and well characterized, it seems that the paper does not quite meet the announced purpose of the study. Thus, the aim of the paper should be changed to better meet the presented results.

Additional comments

- lines 75, 123, 137, 138: Is "B. funghomi" correct or should it be B. funghomii? Please check. Also, the genus and species names of plants should be italicized, this also applies for Chenopodium quinoa (lines 127 & 133), Nicotiana benthamiana (lines 127, 133 & 137) and Dendrocalamus latiflorus (lines 226 & 308).

- line 128: specify which protein was detected by western blot (CP or TGBp1? both are described in the cited paper).

- lines 133-135 and figure 1: Although it is interesting to compare the new BaMV isolates to the reference strain BaMV-S, they should also be compared to their parental virus (BaMV-VN1 & BaMV-VN2), hence one expects to see the symptoms induced by sap inoculation of the parental viruses on C. quinoa and N. benthamiana plants (if mechanical inoculation with bamboo sap is possible).

- Lines 153-167. Sequence identities are given in percentage of the full-length sequence. See lines 152-153: "the Vietnam isolates of BaMV are very distinct from the BaMV strains/isolates from other regions in the world, sharing only 76-78% sequence identities". However, the species demarcation criteria in the genus Potexvirus are defined for the CP or TGBp1 protein sequences (lines 163-163). Thus, the identities between the new clades BaMV-VN and the others, especially the clade BaMV-MT should be given as percentage in the CP and TGBp1 ORFs.

- Legend of Fig3: Please replace GenBank number with virus name that is MT111579.1 with BaMV-Yoshi2 and MH423501.1 with AltMV.

- Lines 190-191: "The fragment corresponding to nts 1395 to 1681 relative to BaMV-VN2-1 was found to be the hybrid origin". Please specify which part of the genome it corresponds to, for example …nts 1395-1681 lying in the 5' half of ORF1.

- Lines 216-217: "Six sequence variants each were selected for satBaMV-VN1 or satBaMV-VN2 for sequence comparison and phylogenetic analysis." Please specify what were the criteria to select these 6 sequences.

- Fig 4: the resolution does not allow to see the satellites nor to read the values on the nodes.

- Table 1: please add a column with virus-isolate names and a column with geographic origin, and highlight outgroup viruses. As this table and the heatmap in fig 5 represent the same results, is it pertinent to keep both in the main part of the paper? Or could the heatmap be changed to contain all information and the table presented as supplementary data?

- Lines 376-379: "Thus consensus primers may be easily designed to amplify this highly variable regions of most BaMV isolates and identify the geographical origins of the BaMV isolates through sequencing, which is analogous to the widely applied technique for the identification of microbes using the 16S/18S ribosomal RNA variable regions". As virus variability/similarity is limited and only make sense within a genus or a family and is therefore very different from microbes, a comparison with other virus classification rather than microbes would be more appropriate.

- lines 84-86: "Then the 5’ primer was synthesized base on the above information" please change base for based.

- line 93: "The nucleotide sequences of BaMV Vietnam isolates and were analyzed" a word seems to be missing after "isolates"

- lines 104-105: "The results of sequence comparisons were visualized by the heat-map generated by the heatmap.2() function natively provided in R" either complete parenthesis after heatmap.2 or remove it.

- line 179: "A BaMV strain found in Fuzhou, China has been shown to a recombinant" a verb is probably missing in the sentence.

- lines 185-186: "namely Plantago asiatica mosaic virus (LC422371), Alternanthera mosaic virus, and Fox-tail mosaic virus (NC_001483)", as recommended by ICTV, remove capital letters from the virus names.

- lines 264-265: "regions other than ORF1 coding do not reflect well with that generated by full-length sequences" remove with.

- line 236 change "rrepresentative" for "representative". Also remove extra spaces in the figure legend.

In the reference list:

- homogenize titles by removing extra capitals in ref 12, 20, 21 & 22

- in ref 3: replace 1248 with 1249-1249, if correct

- in ref 8: replace 2014 with 2015

- in ref 9: only keep one e108015

- in ref 10: replace 886-886 with 886 as this represents the article number, not a page number

- in ref 18: replace 208-208 with 208, it represents the article number, not a page number

- in ref 20: replace A., B. with Bleasby, A.

- in ref 23: replace 4673-80 with 4673-4680.Also indicate complete end page in ref 27, 28 and 29.

- in ref 31: replace Consortium, I.R. with ICTV Report Consortium. Also replace 699-700-700 with 699-700.

- in ref 35: check all authors, Chang, T.-Y. seems to be missing.

- in ref 36: Check title, replace Bamboo mosaic with Virus diseases of bamboos (Bambusa spp.) and replace 723-724 with 723-724 if appropriate.

Reviewer 2 Report

I thank the authors for putting together this manuscript. I would recommend, however, to revise the manuscript and change it to a “Disease Note” or “Brief Report”.

All results are based only in two (!) samples which is by far not enough to make claims about genetic diversity, molecular evolution and epidemiology!

Authors use the words “isolate” and “strains” as it was the same thing. This must be changed! These are vey different terms and should be properly defined.

Introduction could be improved, mainly to provide information regarding Satellite RNAs and a better description of the function of the different ORFS, mainly the TGB. As mentioned above, the number of samples tested are not enough to support the sentence on line 63.

Material and Methods section must be improved. It must provide more information and a clear description of the methodology to support the results.

Results need more clarification:

- Why not use PCR primers for the initial detection of BaMV?

- The 3 closely related virus (lines 185-186) are found in bamboo?

- How can these viruses be a source of recombination?

- The recombination event found was supported by how many methods implemented in the RPD4? where are the results?

- why not building phylogenies with all proteins instead of using full-genomes? why using full-genomes to check BaMV diversity?

- Figure 5: the pattern obtained with TGB1p seems very similar to ORF1. why not use this region for phylogeny? could this protein be a potential target to a genetic marker?

- there are no currently regions to design diagnostic PCR primers?

- authors could perform selection pressures (dN/dS) analysis in all proteins to support the results on section 3.5. These data would avoid speculation as stated on line 359.

Line 379: this comparison is not true and must be removed. For that authors should use general Alphaflexiviridae or Potexvirus primers.

In conclusion, I would recommend re-writing the manuscript and re-submit it as a Disease Note” or “Brief Report”.

Reviewer 3 Report

In this study, the authors found the new isolates of bamboo mosaic virus (BaMV) and new satellite RNAs from bamboos in Vietnam. Phylogenetic analyses on the full-length BaMV and satBaMV RNAs revealed that the Vietnam isolates of BaMV and satBaMV RNAs are distinct from all known BaMV and satBaMV RNAs. The recombination detection analysis revealed that possible recombination with Alternanthera mosaic virus was detected in the genome of BaMV-VN2-1 corresponding to nts 1395 to 1681.  The authors identified the highly variable region of BaMV isolates/strains in the ORF1. The similarity patterns generated by ORF1 coding sequence reflected well with that generated by full-length sequences. They discussed that a highly variable region might serve as the genetic marker for geographic or host origins.

Although the discovery of the new BaMV isolates is interesting, there are significant problems with this paper in the following points.

Major comments

(1) I do not understand the concept of the identification of a genetic marker for the classification, which is one of the main topics of this paper. The authors identified the highly variable region of ORF1 as the most similar region to the molecular phylogenetic tree using the full-length BaMV sequence. They claimed that region might be the genetic marker for identifying the origins of BaMV isolates for epidemiological analyses. However, it is known that BaMV isolates/strains are recombinant, and ORF1 is included in the recombinant region. It has been reported that molecular phylogenetic trees using different gene regions of BaMV show different tree shapes [5, 11]. First, the authors should demonstrate the validity of using full-length sequences for the classification of BaMV in which recombinants frequently occur. In addition, the highly variable region identified by the authors corresponds precisely to the recombinant region with Alternanthera mosaic virus. The deletions and insertions in the BaMV-VNs seen in Fig. 6 may be due to recombinant. Is it reasonable to use such a region as a marker just because it resembles a full-length phylogenetic tree? To examine these and discuss the classification of BaMV, molecular phylogenetic analysis by using all ORFs should be performed. 

(2) Most of the previous papers on molecular phylogenetic analysis of BaMV used outgroups, but this paper did not. It is necessary to explain why outgroups were not used in Figs. 2 and 7.

Minor comments

・The scientific name of plants should be italicized.

・L60: [7. 10] →[6, 7, 10]

・L75-76: designated BaMV-VN1 and BaMV- 75 VN2, →Naming a virus on bamboo is wrong.

・L105: heatmap.2()?

・L109: the main differences→What is the main differences?

・L128: the samples→What was this sample? Bamboo?

・L163-167: →The sequence similarities for each ORF has not been presented here yet; perhaps Table 1 should be given here.

・L185-187: Plantago asiatica mosaic virus (LC422371), Alternanthera 251 mosaic virus (MH423501), and Foxtail mosaic virus (NC_001483)→The abbreviation of virus name should be added.

・L206: vaious→various?

・L207-218: →Move to Materials and Methods

・L230-233: satBaMV RNAs currently identified in Vietnam may have different origins, and provided further support that the satBaMV RNAs may co-evolve with their helper virus. These observations also demonstrated that satBaMV RNAs are widely associated with BaMV from different geographical origins. →In the molecular phylogenetic tree, satBaMV RNAs make a clade, so it can be considered the same ancestor, right?

・L251-252: Plantago asiatica mosaic virus (LC422371), Alternanthera 251 mosaic virus (MH423501), and Foxtail mosaic virus (NC_001483)→Use abbreviations.

・L358-361: This observation suggested that this highly variable region is strongly affected by the evolutionary pressure, and that the sequence variations may have main contributions to the fitness of BaMV in specific environment.→Shouldn't you do a proper population genetics analyses with BaMV isolates/strains to do such discussion?, such as nucleotide diversity (π), average of nucleotide differences per site,  average number of even differences by synonymous (dN), average number of even differences by nonsynonymous sites (dS), dN/dS ratio {ω(dN/dS)}, and Tajima's D test, etc.